# Caffeine Health Claims on Sports Supplement Labeling. Analytical Assessment According to EFSA Scientific Opinion and International Evidence and Criteria

**DOI:** 10.3390/molecules26072095

**Published:** 2021-04-06

**Authors:** Pedro Estevan Navarro, Isabel Sospedra, Alejandro Perales, Cristina González-Díaz, Rubén Jiménez-Alfageme, Sonia Medina, Angel Gil-Izquierdo, José Miguel Martínez-Sanz

**Affiliations:** 1Faculty of Health Sciences, University of Alicante, 03690 Alicante, Spain; pedroestevandn@gmail.com (P.E.N.); rja10@gcloud.ua.es (R.J.-A.); 2Nursing Department, Faculty of Health Sciences, University of Alicante, 03690 Alicante, Spain; isospedra@ua.es; 3Communication Sciences and Sociology, Faculty of Communication Sciences, Rey Juan Carlos University, 28933 Madrid, Spain; aperales@auc.es; 4Psychology and Social Communication Department, Faculty of Economics and Business, University of Alicante, 03690 Alicante, Spain; cristina.gdiaz@ua.es; 5Quality, Safety, and Bioactivity of Plant Foods Group, Department of Food Science and Technology, CEBAS-CSIC, University of Murcia, 30100 Murcia, Spain; smescudero@cebas.csic.es

**Keywords:** nutrition, sport, caffeine, performance, health claims, fraud

## Abstract

Caffeine is a food supplement widely consumed by athletes, but it has not been established. So far, the veracity of their labeling in terms of the dosage and cause/effect relationship aimed at the consumer. The aim is to analyze the health claims and the dosage presented on the labeling of caffeine supplements and to evaluate if they follow the European Food Safety Authority (EFSA) and international criteria. A descriptive cross-sectional study of a sample of caffeine supplements was carried out. The search was done through the Amazon and Google Shopping web portals. In order to assess the adequacy of the health claims, the guidelines of reference established by European Food Safety Authority were compared to the Academy of Nutrition and Dietetics, International Olympic Committee, and Australian Institute of Sport guidelines; in addition, recent systematic reviews were addressed. A review of labels of 42 caffeine supplements showed that, in less than 3% of the products were the health claims supported by the recommendations and by the labeled quantity of caffeine. The claims that fully complied the recommendations were, “improves or increases endurance performance”, “improves strength performance”, or “improves short-term performance”. In most cases, the recommended dosage was 200 mg/day for these products, which is the minimum for the caffeine effects to be declared. The rest of the health claims were not adequate or need to be modified. Most of the health claims identified indicated an unproven cause and effect, which constitutes consumer fraud, and so must be modified or eliminated.

## 1. Introduction

In the field of sports, whether at an amateur or professional level, the use of sports food supplements (SFS) by athletes is increasingly common. The International Olympic Committee (IOC) defines them as “a food, component, nutrient or non-food component that is purposely ingested within the normal diet with the objective of obtaining a determined effect on health or performance” [1].

There is currently a high intake of SFS by the population of athletes [2,3]; 40–100% of athletes of any level consume them [4] and constitute the main commercial target of an industry [5] that offers for sale several thousand different products in this category [4,6]. This industry employs very aggressive marketing and, in particular, commercial communication strategies to achieve a high volume of sales, alleging that SFS improve sports performance (SP) or making other health claims not directly related to the SP [5,7]. Among the most consumed SFS, alone or as part of other sports supplementation preparations, is caffeine [8], one of the supplements whose use is backed by scientific evidence, along with creatine, nitrate, sodium bicarbonate, and β-alanine [1,9,10]. In addition, scientific organizations and public institutions, such as the European Food Safety Authority (EFSA), have previously studied the characteristics, safety, and dosage of caffeine consumption [11].

The commercial communications and labeling must obviously be subjected to scientific scrutiny of their veracity before approval of the consumption of SFS (there must be a defined effect and cause–effect relationship, described by the EFSA in its terms of reference, or TOR). They are also the subject of transversal and specific legislation in the matter of commercialization, advertising, and the truthful and complete transmission of the information of the products that, as in this case, are intended for consumption by the public. At the European level, the SFS are covered by Regulation (EC) 1924/2006 regarding nutritional and health claims for food; Regulation (EU) 1169/2011 on food information provided to the consumer; Regulation (EC) 258/97 on novel foods and novel food ingredients; and Directive 2002/46 on the approximation of the laws of the Member States relating to food supplements [12].

Caffeine is an alkaloid of the family of methylated xanthines and is found in various foods of plant origin, such as coffee, cocoa beans, tea leaves, guarana berries, and kola nuts [11]. In addition, it is an ingredient that is added to a wide variety of foods, such as SFS in capsule, vial, pill, or chewing gum format, and hydrating drinks intended for use by sportsmen and -women [1,11,13]. There are several mechanisms by which the ergogenic effect attributed to caffeine can occur. It acts as an antagonist of adenosine receptors, increases the release of adrenaline and endorphins (thereby leading to a possible lower perception of pain), increases neuromuscular activity and contraction due to greater intracellular calcium release, increases alertness, and reduces the perception of effort and fatigue during exercise. Properties related to the management of weight control and anti-inflammatory effects have also been described for caffeine ingested in coffee [11,14]. Conversely, excessive consumption of caffeine can cause different adverse effects, such as addiction, gastrointestinal upset, nausea, diarrhea, headache, hypertension and tachycardia, mental confusion and lack of concentration, nervousness, and anxiety [1,11,15,16]. The dosage and timing recommendations for caffeine in sport are 3–6 mg/kg of body weight 60 min before exercise [1,10]. Doses greater than 6 mg/kg of body weight have not been shown to give improvements in performance, and for doses greater than 9 mg/kg of body weight, the risk of adverse effects is increased [17].

Caffeine is one of the few SFS for which the health claims are looked on favorably by the EFSA, although they are not approved by the European Commission. The EFSA-recognized health claims are, “improves endurance sports performance”, “increases endurance”, and “reduces perceived exertion during exercise”, provided a minimum caffeine content of 70 mg per dose of product is guaranteed [11,18]. These health claims occupy a central position in the marketing of these products, both in their labeling and in the advertising intended for athletes [19]. However, research in this area has detected cases in which the health claims do not entirely match the described effects that some food supplements produce on the health of the users who consume them [20].

The aim of this study was to analyze the information provided in the health claims and dosage present in the labeling of caffeine food supplements, as well as to verify their degree of compliance with the regulations and to check the adequacy of these health claims, according to the EFSA scientific opinion and current scientific evidence in the European context.

## 2. Results

In the search, 414 results were obtained, of which 42 caffeine supplements belonging to different trademarks met the inclusion criteria established in the methodology. Hence, 372 results were rejected: 327 for not meeting the format and/or composition criteria; 11 because the caffeine content did not appear on the label; 29 for appearing repeatedly on one or both of the commercial portals; and 6 for stating insufficient doses, not presenting health claims on the official website, or not meeting the established criteria (Figure 1). For the 42 selected supplements, the health claims presented and the dosage of the supplement are specified.

### Health Claims, Product Dosage, and Compliance with Current Scientific Evidence

As can be seen in Table 1, regarding the caffeine dose of the products, 2.44% (*n* = 1) of them contained 70–99 mg per dose, 7.32% (*n* = 3) contained 100 mg per dose, 4.88% (*n* = 2) contained 101–199 mg per dose, and 85.36% (*n* = 35) contained 200 mg per dose. In addition, Table 1 establishes a percentage distribution of each healthy property declaration found in the SFS sample and further compares the proposed dosages and the type of healthy property declaration indicated by the manufacturer in each caffeine sports supplement. The health declaration most frequently described for the caffeine SFS was, “gives energy” and “improves concentration/cognitive improvements,” found in 56.1% of the sample, followed by, “improves focus/alertness” (51.2%), “reduces physical/mental fatigue” (41.5%), and then, in turn, “performance improvement” and “reduces physical/mental fatigue”.

There were also health claims that only appeared in a single product from the sample of sports supplements, such as, “improves the immune system,” “improves coordination,” and “optimizes the use of glycogen reserves.” Each of these represented 2.4% of the health claims of the total sample.

In the case of the health claim, “reduces the perceived effort,” minimum doses of both 70 mg and 200 mg were indicated; as well, for, “increases resistance performance,” these two doses featured in various products that declared this effect of caffeine.

The health claims, “increases performance in the short term,” “improves performance by improving strength,” “stimulates the central nervous system (CNS)”, “increases alertness,” and “improves neuromuscular function,” were recommended in most supplements at a dosage of 0.3 mg/kg (approximately 200 mg per intake) of the caffeine product.

## 3. Discussion

In the present study, the health claims made on their labeling for a sample of caffeine-based products have been analyzed, as well as the dosages indicated to achieve these effects. We found that the most commonly recommended dosage (for 85.36% of the products of the sample) was 200 mg per day, followed by the dose of 100 mg per day (7.32%), with the rest of the doses having an even lower presence.

The health claims present most frequently among the caffeine products/supplements (in 56.1% of them) were, “gives energy” and “improves concentration/cognitive improvements,” followed by “improves focus/alertness” (51.2%) and “reduces physical/mental fatigue” (41.5%).

The use of caffeine supplements in sport forms part of the search for an improvement in physical performance in various sports modalities. There are many brands that market these supplements, the effects they produce, and the manufacturer′s recommended dosage featuring on the label. This dosage usually coincides with the dose present in a single intake of the supplement (capsule, pill, vial, etc.). In some cases, these data do not comply with what has been established by various institutions—such as the EFSA, American Academy of Nutrition and Dietetics (AND), IOC, and International Society of Sports Nutrition (ISSN)—or in the latest reviews of scientific evidence [1,9,10,11,21,22,23,24].

### 3.1. Health Claims and Proposed Dosages

Currently, athletes are exposed to a significant volume of commercial communications that attribute performance improvements to the promoted products, including SFS. Authorities and consumers should insist that these allusions and advertisements regarding nutritional and health claims, as well as claims of ergogenic effects, be supported by scientific evidence and not confuse/mislead consumers by exaggeration of the ability of a certain product to improve performance [5,25,26]. Athletes often obtain little and/or erroneous information about the uses and functions of SFS, and, therefore, their intake must be supervised by a professional. In addition, some studies have shown a deficiency in the nutritional knowledge of athletes in aspects related to both general nutrition and the specific needs of sports practice [7].

This lack of knowledge may also be due to, and even aggravated by, erroneous and unfounded beliefs about eating habits shared by friends, family, coaches, advertising, etc. [5,27], to which credibility is granted.

In this study, there were a total of 25 different health claims presented in the sample of supplements by the manufacturers, of which only 8 were totally or partially in accordance with those established by institutions, consensus documents, or scientific evidence. These health claims are approved at the European level by both the EFSA and the European Commission (EC) [28]; in addition to establishing specific legislation to allow the regulation of sports nutrition products and their advertising through consensus documents, there is a record of the health claims made in the marketing of these specific products [11,12,29].

Only the health claims, “increases strength performance,” “increases performance in the short term,” “increases performance in endurance sports,” “improves performance,” “decreases tiredness/feeling of fatigue,” “increases concentration,” and “increases focus/alertness” can be considered to satisfy the criteria of the leading scientific institutions [1,10,11,17,21,22,23,24]. In addition, it should be noted that the effects declared for caffeine relate to a stated minimum dose of 200 mg/dose and day of the product.

### 3.2. Fraud in Advertising and Direct Consumer Information

As has been observed, the advertising of a product does not always refer correctly to the effects of a particular food. Food fraud can be found in various guises within the advertising, consumer information, and marketing of SFS.

In addition to incorrect data based on health claims, dosing errors can also be found, as well as differences and errors in the labeling and composition of a product. Prohibited or undeclared substances may even be present, an aspect of importance for the World Anti-Doping Agency (WADA) and which can pose a serious risk to the health of consumers [30,31,32].

### 3.3. Action and Proposals to Deal with Incomplete, Inaccurate, or Confusing Health Claims

From the point of view of the advertising and marketing of food products, the EFSA is involved in food safety in the context of public health at the European level. In addition, regulation through legislative documents serves as a legally binding instrument against advertising and food fraud. In each EU Member State, the current regulations that enforce the legislation and apply it to ergogenic nutritional aids or sports nutrition products include the following Regulations: (EU) 1169/2011, on the food information provided to the consumer; (EC) 353/2008, on requests for authorization of healthy property declarations; (EC) 1924/2006, on nutritional and health claims made for food; (EC) 1925/2006, on the addition of certain substances to foods; and (EC) 258/97, on novel foods and novel food ingredients, as well as Directive 2002/46/EC, concerning the approximation of the laws of the Member States regarding food supplements [12]. To this must be added the transversal regulations—generally also of European scope, regarding their direct application, or incorporated into each national legal system—which regulate advertising legality and good commercial practices, in general or for certain media and communication channels.

All this legislation places special emphasis on the need for communications to be truthful, not to mislead users, and, in matters of health, to be subject to scientific evidence and to the allegations authorized by the health authorities. In the case of food, SFS, and the associated advertising, the criteria established by the WADA must also be taken into account for the presence of prohibited substances in food [30].

In Spain, to regulate food advertising, in addition to the work of the competent health and consumer authorities and the work of the courts, the advertising and media sector has an independent body for extrajudicial conflict resolution at the national level, the Association for the Self-regulation of Commercial Communication (Autocontrol). In addition, there are organizations, such as the Association of Communication Users (AUC), which is dedicated to defending the rights of citizens as users of different media and communication systems, offering any user the possibility of reporting any advertising content they consider unlawful. In this way, the AUC can instigate actions against said commercial communications in any of the three previously mentioned areas: administrative, judicial, or voluntary regulation.

### 3.4. Advertising Fraud Cases

Fraudulent food-related situations often occur, as mentioned above. In the “Foodwatch” study carried out in the Netherlands and Germany, claims regarding health effects and the presence of high concentrations of certain nutrients as a marketing tool for unhealthy foods stand out [33].

There are several studies that show fraud in the labeling of supplements, especially related to proteins, creatine, or weight loss; this could be due to involuntary or voluntary adulterations or other contamination [34,35], making necessary, as some authors indicate, a regulation that seems “mission impossible” [36].

It has been found that many products do not present the scientific references correctly or do not show them directly, which indicates that the information that reaches the athlete is—in many cases—scarce, erroneous, or at least confusing, in terms of general and sports nutrition; this may affect their eating habits and performance counterproductively [7]. In this regard, Molinero et al. pointed out that 52.8% of the supplements they analyzed did not present scientific references referring to the health claim attributed to them [37].

There are also studies that show that the caffeine content stated on the label does not correspond to the actual content of this substance in the supplement, with considerable differences in more than 50% of the supplements analyzed [38,39,40]. This is especially common for supplements that contain caffeine as the main ingredient, such as preworkout SFS. In one of these studies, it was shown that only 6 out of 15 supplements analyzed included details of the caffeine content on the label, and that the true content was between 59% and 176% of that declared in the nutritional panel [40]. This type of negligence is not exclusive to SFS; it has also been evidenced in other types of products, such as dietary supplements and food. [41]. In our results, we observed a low adequacy rate of the health claims made by the manufacturer in relation to those established by the reference institutions and the scientific literature. Health claims that are inadequate or do not correspond to the product in question can encourage the purchase of the product by consumers who seek the benefits indicated by the manufacturer, meaning that they are deceived.

Therefore, it is vital that the authorities demand truthful, high-quality, and safe advertising, supported by up-to-date scientific evidence, from the brands that use these health claims, in order to increase their sales by confusing the consumer. This would guarantee legitimate and truthful advertising for the products in question [42].

### 3.5. Study Limitations

One of the limitations of the study is the heterogeneity of the results of the search portals, as well as the existence of products that did not offer the information required for the study. This work also highlights the multitude of health claims offered by manufacturers or advertisers, in some cases representing very confusing information that increases the complexity of the analysis. In addition, several SFS had unclear labels or errors in typology and dosage, so they were left out of the screening, even though they may have been valid.

On the other hand, this study is focused in the European context (EFSA scientific opinion) and the scientific evidence and criteria provided by literature and international institutions. Future research could extend this work. Concretely, it could analyze and compare European and US context about this issue.

## 4. Materials and Methods

### 4.1. Type of Study

An observational and cross-sectional study was performed based on analysis of the different health claims included in the commercial communications of a sample of caffeine supplements, analyzing these claims in the light of EFSA scientific opinion and scientific evidence in this area, following the methodology from a previous research [43].

First, we reviewed the EFSA scientific opinion papers, based on this organization′s TOR, and scientific research in this area. Second, we analyzed the contents of the different health claims included in the commercial communications of a sample of caffeine supplements. Finally, we evaluated whether these health claims are in accordance with the scientific documents reviewed.

### 4.2. Study Population Selection Strategy

The search for the sample products was carried out in March 2020 through the Amazon and Google Shopping websites. On each of these websites a filter was applied (country or European region) to carry out the search process and to obtain results for caffeine supplements. To carry out the search process, the term “caffeine” was introduced in both portals. In the Amazon portal, filters were added for the “diet and nutrition” and “sports supplements” departments, while in Google Shopping, a filter was added with the terms “pill” and “capsule”; then, “caffeine vial” and “caffeine gum” were individually searched. From this initial search, supplements that contained caffeine as the single ingredient (pills, capsules, liquid vials, or chewing gum forms) were selected. Once the sample had been obtained, each of the web portals of the selected supplement brands (the company′s website) was visited to observe the health claims for each of them (see Appendix A). The process of obtaining each component of the sample was different depending on the portal visited.

### 4.3. Inclusion Criteria

Caffeine supplements in “capsule,” “pill,” “vial,” or “chewing gum” forms, with caffeine as the single ingredient, and for sale in Europe formed part of the selected sample. In addition, the caffeine supplements obtained from the selected web portals that declared some beneficial property regarding the athlete′s sports performance or health, as well as those that provided information on the dosage of the product, were included in the sample.

### 4.4. Exclusion Criteria

Supplements containing caffeine but combined with other ingredients were excluded from the sample. Furthermore, supplements for which no health claims were declared or for which information on the dosage of the product was not provided were also not part of the selected sample. The repeated samples were also excluded.

### 4.5. Data Extraction

After carrying out the search to select the study sample, a descriptive analysis of the characteristics of each selected caffeine product given in the labeling was performed. The variables studied for each product in the sample were as follows:Product name: the name of each of the SFS belonging to the study sample was specified.Dosage: the consumption (amount) recommended by the manufacturer for each SFS.Health claims related to caffeine: those present in the labeling of each of the SFS of the selected sample.Degree of compliance: this refers to the extent to which the health claims made in the selected sample of SFS regarding caffeine matched the health claims (effect defined according to the EFSA TOR), regarding this substance approved by current scientific evidence (cause–effect relationship according to the EFSA TOR).Health claims modification: removal/modification of the health claims made on the labeling of the caffeine supplements so that they are in accordance with current scientific evidence.

### 4.6. Compliance with Legislation and Scientific Evidence

The commercial communications and the labeling of the selected sample were evaluated through the effect and the cause–effect relationship, defined by the EFSA in its TOR. In addition, the results were also compared in relation to the scientific evidence and criteria of the International Olympic Committee (IOC) [1], the American Academy of Nutrition and Dietetics (AND) [10], and the Australian Institute of Sport [17], as well as the most recent scientific documents on caffeine, such as systematic updates and meta-reviews analysis [21,22,23,24] (Table 2).

## 5. Conclusions

The health claims that fully complied with the recommendations are, “improves performance in the short term” and “improves performance in resistance” (2.78% of the total). Only 32 health claims (30.7% of the total) were partially adequate (score of 3 or 4), because these lacked the specification of sport and timing. The health claims made for SFS should conform to the criteria established by EFSA opinion and scientific evidence. Sports foods supplements fraud is found in various forms within the advertising and marketing of food, strongly affecting consumers. For this reason, it must be improved public health or sports policies and European regulation applied directly to sports foods supplements.

## Figures and Tables

**Figure 1 molecules-26-02095-f001:**
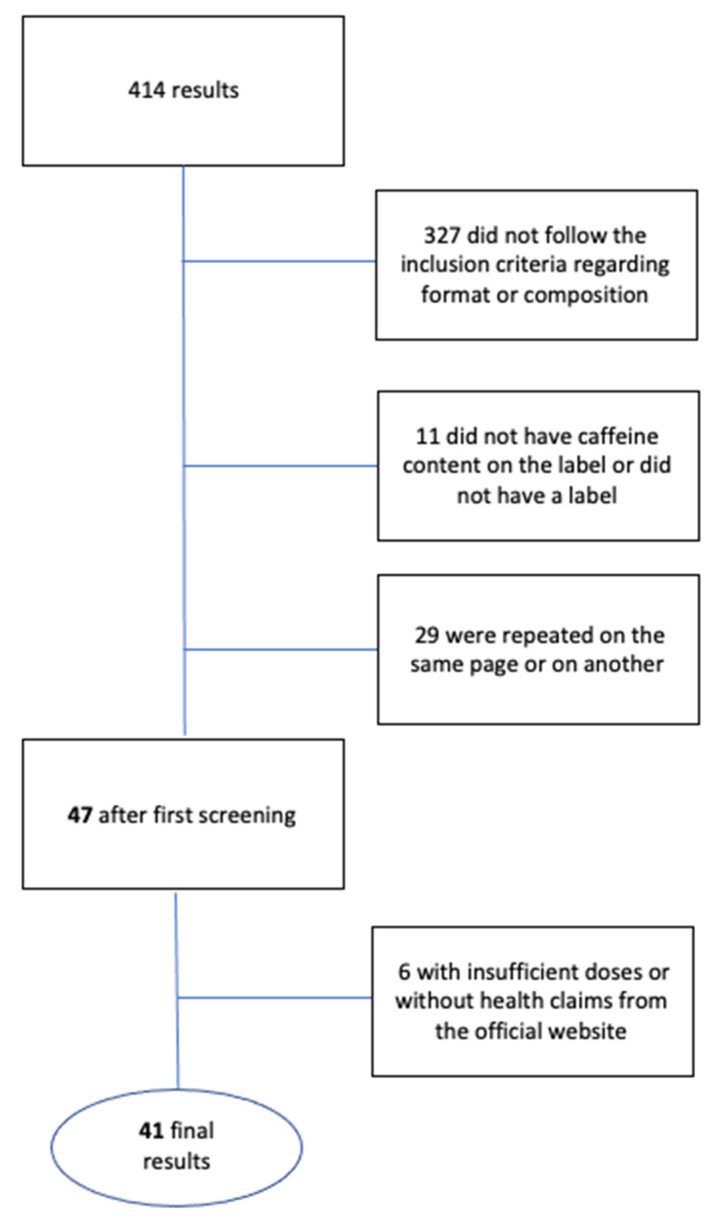
Flow diagram showing how the study sample was obtained.

**Table 1 molecules-26-02095-t001:** Distribution of caffeine supplement doses according to health claims and their reasons of adequacy with European Food Safety Authority (EFSA) scientific opinion.

Health Claims	Number and Percentage (%) of Total Supplements for which this Health Claim is Made	Dose (mg/Serving)	Number and Percentage (%) of Supplements for which this Dosage is Given for the Health Claim(s) Stated	Reason for Lack of Adequacy *
Reduces physical and/or mental fatigue	17 (41.5%)	101–199	1 (5.9%)	2
		100	2 (11.8%)	2
		200	14 (82.3%)	2
Increases endurance performance	10 (24.4%)	70–99	1 (10%)	5
		200	9 (90%)	4 (*n* = 6)
				5 (*n* = 3)
Energizes	23 (56.1%)	70–99	1 (4.35%)	1
		100	1 (4.35%)	1
		101–199	1 (4.35%)	1
		200	20 (86.95%)	1
Thermogenic action	5 (12.2%)	200	5 (100%)	1
Diuretic action	5 (12.2%)	200	5 (100%)	1
Increases concentration/cognitive improvements	23 (56.1%)	70–99	1 (4.3%)	3
		100	2 (8.7%)	3
		101–199	2 (8.7%)	3
		200	18 (78.3%)	2
Improves performance	16 (39%)	100	3 (18.75%)	3
		200	13 (81.25%)	3
Stimulation	14 (34.1%)	101–199	2 (14.3%)	2
		200	12 (85.7%)	2
Improves memory	3 (7.3%)	200	3 (100%)	1
Improves metabolism	6 (14.6%)	200	6 (100%)	1
Aphrodisiac	2 (4.8%)	101–199	2 (100%)	1
Enhances power/strength performance	2 (4.8%)	200	2 (100%)	4
Improves short-term performance	4 (9.8%)	70–99	1 (25%)	5
		200	3 (75%)	4
Muscle recovery	1 (2.4%)	101–199	1 (100%)	1
Oxidizes/burns fat	8 (30%)	100	1 (12.5%)	1
		101–199	1 (12.5%)	1
		200	6 (75%)	1
Improves coordination	1 (2.4%)	200	1 (100%)	2
Optimizes the utilization of glycogen reserves	1 (2.4%)	100	1 (100%)	1
Enhances the immune system	1 (2.4%)	200	1 (100%)	1
Improves focus/alertness	17 (51.2%)	100	2 (11.8%)	2
		200	15 (88.2%)	2
Treatment of pathologies	2 (4.8%)	101–199	1 (50%)	1
		200	1 (50%)	1
Helps weight loss	11 (26.8%)	100	1 (9.1%)	1
		101–199	2 (8.2%)	1
		200	8 (72.7%)	1
Helps breathing	1 (2.4%)	200	1 (100%)	1
Increases heart rate/vasoconstriction	2 (4.8%)	200	2 (100%)	1
Lowers fat%	2 (4.8%)	200	2 (100%)	1
Polyphenols/antioxidants	3 (7.3%)	101–199	1 (33.3%)	1
		200	2 (66.7%)	1

*Source: Own elaboration based on the search data.** Reason according to EFSA scientific opinion: Number 1. Cause: it does not conform to the approved health claims regarding caffeine. Modification proposal: delete product health claims; Number 2. Cause: it conforms to the approved health claims but does not provide the recommended dosage of the product. Modification proposal: change product dosage protocol; Number 3. Cause: it conforms to the approved health claims and the correct dose established but does not specify the type of exercise performed. Modification proposal: change health claim, specifying the type of exercise in which the claimed effects are shown; Number 4. Cause: it conforms to the approved health claim, specifying the appropriate recommended dose of the product and the effects of the supplement, but not the timing. Modification proposal: change the health claims, specifying the timing; Number 5. Cause: it fits all the above and includes the intake timing. Modification proposal: do not modify or remove the health claims.

**Table 2 molecules-26-02095-t002:** Effects and applications of caffeine established by the EFSA scientific opinion and the scientific evidence and criteria provided by the literature and International Institutions.

	Types of Caffeine	Sports Involved	Dose	Ergogenic Effects
**EFSA Scientific opinion**				
EFSA [11]	Does not specify type of caffeine.	Endurance sports.	Minimum 70 mg/dose of product. 3 mg/kg body weight (200 mg approximately), 1 h before physical exercise. * 4 mg/kg body weight to reduce the perception of effort during exercise.	Increases endurance performance. Increases resistance capacity. Reduces the perception of effort during exercise.
**Scientific evidence**				
IOC [1]	Anhydrous caffeine (pills or powder).	Endurance sports, intermittent sports, and short-term sprints sports.	3–6 mg/kg body weight, 60 min before exercise. >3 mg/kg body weight before or during exercise, accompanied by a source of CHO.	Improves neuromuscular function. Heightens the state of vigilance and alertness. Reduces the perception of effort during exercise.
AIS [17]	Caffeine.	Endurance sports, team/intermittent sports, high intensity physical activity of short duration.	At 1–3 mg/kg body weight, benefits are observed. At >6–9 mg/kg body weight, there are no benefits. *	Directly contributes to optimal performance. Stimulates the CNS. Decreases the perception of effort during exercise. * Variability between subjects *
AND [10]	Does not specify the type of caffeine.	In its bibliographic references: endurance sports, intermittent sports, or team sports.	3–6 mg/kg body weight.	Acts on the CNS, reducing the perception of fatigue. Promotes the release of Ca^2+^ in the endoplasmic reticulum of skeletal muscle.
Systematic review and meta-analysis [21]	Caffeine.	Endurance sports.	3–6 mg/kg body weight.	Improves sports performance.
Systematic review and meta-analysis [23]	Caffeine (capsule or liquid form).	Sports of muscular strength and power (vertical jumps).	0.9–7 mg/kg body weight.	Improves sports performance through greater application of muscle strength and power.
A meta-analysis [24]	Caffeine (capsule or beverage form).	Endurance sports.	3–6 mg/kg body weight	Improvement of sports performance (higher concentrations of blood glucose and blood lactate and lower perception of effort during exercise).
Systematic review and meta-analysis [22]	Caffeine (any dose and form).	Strength sports	1–9 mg/kg body weight or 150–328 mg	Improves the speed of movement in strength exercises, with medium and moderately high loads

EFSA: European Food Safety Authority; AND: Academy of Nutrition and Dietetics; AIS: Australian Institute of Sport; IOC: International Olympic Committee; CNS: Central Nervous System; Source: Own elaboration based on data collected from different institutions.

## Data Availability

The data presented in this study are available in the tables and Appendix A of this article. The data presented in this study are available on request from the corresponding authors.

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
