# Peer review of "Caffeine Health Claims on Sports Supplement Labeling. Analytical Assessment According to EFSA Scientific Opinion and International Evidence and Criteria"

_molecules, 2021, doi:10.3390/molecules26072095_

Round 1
Reviewer 1 Report
The authors review labeling on available caffeine products marketed for sports in Europe and report on accuracy of claims in relation to quantity of caffeine. Their work is scholarly and thoughtful, but would be strengthened by addressing several important issues.
Please be more precise about the nature of the products studied. The description of “Supplements that contained only CF as active ingredient…were selected” is confusing. Did that authors only select supplements that contained a single ingredient, i.e., caffeine? Or did they review products that contained any number of ingredient one of which was caffeine? Did they also have caffeine-only products included? How many of each category? Also what does “active” in this sentence mean? When would caffeine be an inactive ingredient?
I find it difficult to understand how all the inclusion/exclusion criteria support the research question. The question the authors are asking seems to be, ‘are claims on caffeinated sports supplements sold in Europe accurate?’ if that’s the case, please state more clearly. also, why exclude caffeinated products that do not list the amount of caffeine but do contain a claim – clearly this claim cannot be supported by the labeled information and should be included in a survey of caffeinated products with inappropriate claims.
On page 10, line 298: it is stated that “in addition” to caffeinated products, products that had health claims on these web sites were included. This is very confusing. So, products without caffeine were also included?
Line 304, page 10: the statement “supplements that were not defined as ‘caffeine’” does not make sense
The description of these products is confusing from the American perspective – in the US caffeine can both be sold in dietary supplements but also as an over-the-counter medication. The regulatory frameworks for these two categories of products are completely different, including the permitted marketing claims. It might be useful for a broad readership to provide a general overview of how caffeine can be sold in Europe and compare this to the US model (briefly).
It might also be insightful to briefly compare the legal options for marketing claims on these products to those permitted in the US which is a much more permissive marketplace (see for example: doi:10.1001/jama.2016.14252).
Abstract is quite weak. Compared to the thoughtfulness of the introduction and discussion, the abstract is not clear, precise or compellingly, nor accurately reflects the study. When it gets to the heart of the findings it states blandly and confusingly, “of health claims identified 2.78% were…in most cases with a recommended dosage of 200 mg/day of these products” this doesn’t seem to mean anything at all stated this way. The authors should revise completely to be crystal clear. For example, a clear statement could be: “A review of labels of 42 caffeinated supplements, in less than 3% of the products were the claims supported by the labeled quantity of caffeine.” - or something to that effect.
Using CF as an abbreviation for caffeine makes the manuscript more difficult to read. Recommend not using this and using caffeine in every occurrence.
Author Response
We attached cover letter to reviewer 1

Reviewer 2 Report
This is a well written paper and a well presented project.
The authors need to modify Table 1, the title at the moment is not well written (i.e. adequacy? Against what? Assessed by whom?).
Also, I would suggest the authors to introduce a separate column in that figure to indicate which claims can be linked to EFSA guidelines.
At the moment the table is very busy for the readers so I would suggest to divide this table to two smaller ones.
Also, English use needs to be improved. There are a number of syntax errors in the MS.
Author Response
We attached cover letter to reviewer 2

Round 2
Reviewer 2 Report
The revised document can now be accepted for publication.